# Intranasal Nanotransferosomal Gel for Quercetin Brain Targeting: II. Antidepressant Effect in an Experimental Animal Model

**DOI:** 10.3390/pharmaceutics15082095

**Published:** 2023-08-07

**Authors:** Mohammed H. Elkomy, Fatma I. Abo El-Ela, Randa Mohammed Zaki, Omar A. Alsaidan, Mohammed Elmowafy, Ameeduzzafar Zafar, Khaled Shalaby, Mohamed A. Abdelgawad, Hany A. Omar, Rania Salama, Hussein M. Eid

**Affiliations:** 1Department of Pharmaceutics, College of Pharmacy, Jouf University, Sakaka 72341, Saudi Arabia; osaidan@ju.edu.sa (O.A.A.); melmowafy@ju.edu.sa (M.E.); azafar@ju.edu.sa (A.Z.); khshalabi@ju.edu.sa (K.S.); 2Department of Pharmacology, Faculty of Veterinary Medicine, Beni-Suef University, Beni-Suef 62511, Egypt; fatma.aboel3la@vet.bsu.edu.eg; 3Department of Pharmaceutics, College of Pharmacy, Prince Sattam Bin Abdulaziz University, Al-Kharj 11942, Saudi Arabia; randazaki439@yahoo.com; 4Department of Pharmaceutics and Industrial Pharmacy, Faculty of Pharmacy, Beni-Suef University, Beni-Suef 62511, Egypt; 5Department of Pharmaceutical Chemistry, College of Pharmacy, Jouf University, Sakaka 72341, Saudi Arabia; mhmdgwd@ju.edu.sa; 6College of Pharmacy, University of Sharjah, Sharjah 27272, United Arab Emirates; hanyomar@sharjah.ac.ae; 7Macquarie Medical School, Faculty of Medicine, Health and Human Sciences, Macquarie University, Macquarie Park, NSW 2109, Australia; rania.salama@mq.edu.au; 8Woolcock Institute of Medical Research, Glebe, NSW 2037, Australia

**Keywords:** intranasal, depression, pharmacodynamics, quercetin, forced swim test, transferosomes

## Abstract

Depression is a serious mental disorder and the most prevalent cause of disability and suicide worldwide. Quercetin (QER) demonstrated antidepressant effects in rats exhibiting anxiety and depressive-like behaviors. In an attempt to improve QER’s antidepressant activity, a QER-loaded transferosome (QER-TFS) thermosensitive gel for intranasal administration was formulated and optimized. The therapeutic effectiveness of the optimized formulation was assessed in a depressed rat model by conducting a behavioral analysis. Behavioral study criteria such as immobility, swimming, climbing, sucrose intake, number of crossed lines, rearing, active interaction, and latency to feed were all considerably enhanced by intranasal treatment with the QER-TFS in situ gel in contrast to other formulations. A nasal histopathological study indicated that the QER-TFS thermosensitive gel was safe for the nasal mucosa. An immunohistochemical analysis showed that the animals treated with the QER-TFS thermosensitive gel had the lowest levels of c-fos protein expression, and brain histopathological changes in the depressed rats were alleviated. According to pharmacodynamic, immunohistochemical, and histopathological experiments, the intranasal administration of the QER-TFS thermosensitive gel substantially alleviated depressive symptoms in rats. However, extensive preclinical investigations in higher animal models are needed to anticipate its effectiveness in humans.

## 1. Introduction

Depression is a mental condition that affects people worldwide (around 264 million) [1]. It is characterized by feelings of anhedonia, a lack of energy, feelings of worthlessness or guilt, a depressed mood, weight loss, impaired concentration, and disrupted sleep or appetite; the degree of symptoms varies from moderate to severe [2]. According to the WHO, depression is the leading cause of disability and suicide [2]. Psychological and physiological stresses produced by stressful life events (e.g., unemployment, poverty, and the end of a relationship) are believed to be key proximal factors to depression [3] and are related to more severe symptoms, higher relapse rates, and longer disease duration [4].

Depression disrupts the hippocampus and prefrontal cortex functions [5]. Both structures are crucial in decision making; consequently, a malfunction at this level may increase the propensity for unpleasant emotions [6]. Persistent stress leads to depression [7], which shows the close link between oxidative stress and mood disturbances [8]. Additionally, a complex pathophysiological process that includes a reduction in the neurotransmission of monoamines (noradrenaline, dopamine, and serotonin) [9], as well as excessive inflammation and immunological changes, are key clinical processes that contribute to the disease [10,11].

The presently available antidepressant drugs (ADDs), which are based on the monoamine theory, take 6–8 weeks to provide measurable results and are helpful for only one-third to one-half of depressed people [2]. Also, ADDs have been associated with serious side effects, including headaches, weight gain, and sexual dysfunction [12]. In treating depression, several treatments, delivery systems, and routes are used today. The efficiency of ADD is contingent upon its prolonged presence in the brain [13]. The BBB inhibits the passage of ADDs administered orally and intravenously into the brain. Thus, there is an increasing need for innovative, more effective, and safe ADD treatments. Today, phytotherapy encourages the use of plants in the treatment of diseases and is a valuable resource in creating more powerful ADDs [14]. Several medicinal plants [15,16] are used to treat neurological illnesses. Owing to their neuroprotective and antioxidant effects, phenolic chemicals such as flavonoids constitute a potential class of naturally occurring molecules relevant to neuropharmacology [17].

Flavonoids are widely distributed in plants; thus, humans eat an average of 1–2 g of flavonoids daily [18]. Quercetin (QER) is the most prevalent flavonoid found in food and consumed by individuals [19]. Numerous studies show that QER has a preventative role against depressive-like illnesses, as it was shown that Ginkgo biloba leaves and freeze-dried onion powder, which are high in QER, possess antidepressant properties [20,21]. Additionally, this flavonoid improves behavioral issues in rats exhibiting depressive-like behaviors that are induced by corticotropin-releasing factors [22,23]. QER is believed to exert its antidepressant activity by reversing oxidative stress processes hallmarked by excessive hydroxide production and the release of free radicals [24,25,26].

Recent interest in nasal medication delivery as a viable route has increased due to its many benefits over oral or parenteral administration [27,28]. Intranasal delivery, in addition to being noninvasive and painless, enhances medication absorption owing to the high levels of vascularization throughout the nasal membrane [28]. Further, the nasal route avoids the hepatic and intestinal metabolisms and enables the administration of medication to the brain both directly via trigeminal neurons and indirectly through the systemic circulation [27]. Recent studies indicates that intranasal administration of medication may be useful for treating a variety of CNS disorders [29,30,31,32]. However, short residence duration in the nostrils due to rapid mucociliary clearance and the restricted surface area of the nasal cavity are considered to be the most significant limitations for intranasal drug delivery [27]. Hence, current study focuses on creating mucoadhesive formulations to maximize nasal absorption and residence duration.

This study is the second part of a two-part investigation. In the first part [33], we optimized quercetin-loaded transferosomes (QER-TFS) for QER delivery to the brain via the intranasal pathway and characterized their size, ζ potential, loading capacity, in vitro release, and ex vivo nasal diffusion. Next, the QER-TFS were incorporated into a mucoadhesive thermolabile gel, and their brain localization potential and cytotoxic considerations were evaluated. The physical parameters obtained for the optimized nanotransferosomes suggest their suitability for intranasal applications with a vesicular size of 171.4 ± 3.4 nm, an entrapment efficiency of 78.2 ± 2.8%, a surface charge of −32.6 ± 1.4 mV, sustained release behavior, and enhanced transnasal permeation. The therapeutic potential of the optimized formulation was substantiated v increased accumulation in the brains of rats and its reduced cytotoxicity in a PCS-200-014 cell line.

In this part of the investigation, the therapeutic effectiveness of the optimized nanotransferosomes was evaluated in an experimental animal model of depression by performing behavioral testing. Furthermore, a c-fos immunohistochemistry analysis and histopathological examination were also conducted. The overall aim of the entire investigation is to deliver QER to the brain in a sustained way via the intranasal pathway to treat depression.

## 2. Materials and Methods

### 2.1. Materials

Quercetin (QER), methanol (HPLC grade), soybean lecithin, sodium deoxycholate (SDC), poloxamer 407 (PF407), chloroform (HPLC grade), poloxamer 188 (PF188), and carbopol 971P (CP971P), were procured from Sigma-Aldrich (St. Louis, MO, USA). A primary polyclonal rabbit antibody against the c-fos protein was obtained from Cambridge Research Biochemicals (Cambridge, UK), CRB OA-11-823, with a 0.1% Triton dilution in PBS at 1:10,000). A secondary anti-rabbit antibody (1:400) was obtained from Vector Labs (Burlingame, CA, USA). All other ingredients and chemicals utilized were of analytical quality.

### 2.2. Methods

#### 2.2.1. Preparation of Quercetin-Loaded Transferosomes (QER-TFS) and Quercetin Solution (QER-SOLN)

The method of preparing the QER-TFS was detailed in the first part [33]. Concisely, the QER-TFS were formulated according to the thin film hydration approach [34,35]. In a round-bottom flask, 200 mg of soybean lecithin, 23.1 mg of SDC, and 10 mg of QER were dissolved in 15 mL of a chloroform/methanol mixture (2:1). The organic solvents were evaporated under vacuum in a rotating evaporator (Stuart rotary evaporator, North Yorkshire, UK) at 60 °C until a thin film was formed. The flask was then kept in a desiccator under vacuum for 2 h to ensure that all traces of the solvents were evaporated [36]. Then, 10 mL of a simulated nasal electrolyte solution (pH of 5.5) was added. The dry film was then completely hydrated by rotating the flask at 60 rpm and 25 °C for one hour. The size of the vesicles was reduced by sonicating the film for 10 min in a bath sonicator [37]. The QER-SOLN was produced by adding QER to a simulated nasal electrolyte solution with a pH of 5.5. Finally, the developed QER-TFS and QER-SOLN were incorporated into a mucoadhesive thermosensitive gel matrix (composed of 20% PF407, 0.5% CP971P, and 10% PF188) that contained 2% *w*/*v* of QER [27,28] for ease of administration to the nasal cavity and minimal dripping. The Diagram showing the scheme for formulation of QER-TFS thermosensitive gel is presented in Figure 1.

#### 2.2.2. In Vivo Antidepressant Activity of QER-TFS Gel

##### Animals

The antidepressant efficacy of the QER-TFS gel was examined using 30 adult male rats that weighed between 160 and 180 gm. The animals were allowed to acclimate to their new environments for a full week before the experiments began.

##### Induction of Depression and Experimental Protocol

The antidepressant efficacy of the QER-TFS gel was investigated using an animal paradigm that generated depressive-like symptoms via the forced swimming test (FST) [38]. Environmental changes may influence the results of behavioral experiments in the FST model of depression. In the pre-test, the rats were generally active, straining to climb the wall, aggressively swimming in circles, or falling to the bottom. The rats were individually left to swim in a vertical glass jar that measured 50 cm in height and 22 cm in diameter and contained 35 cm of water that was kept at a temperature of around 25 °C. The rats were put through a 15 min pre-test swim exercise, followed by a 6 min test swim the next day. After that, each test was completed in six-minute sessions twice a day, for seven days. The rats were removed from the water at the end of each swim session, toweled off, and then returned to their cages. In addition, the jars were depleted of their contents and thoroughly cleaned. This testing session was carried out daily for a total of seven days. Movement tends to diminish after 2–3 min, resulting in a characteristic behavior known as immobility in which the rat makes just the minimum number of movements required to maintain its head above water.

After seven days of the FST, the groups were split up, and each group received a different formulation of QER. The rats were randomly divided into five groups (*n* = 6): the negative group (rats that were not exposed to the FST and were given normal saline), the positive group (rats that were exposed to the FST and did not receive any treatment), the QER-SOLN oral group (rats that were depressed via the FST and were treated with oral QER-SOLN), the QER-SOLN gel (IN) group (rats that were depressed via the FST and treated with the intranasal QER-SOLN gel), and QER-TFS gel (IN) group (rats that were depressed via the FST and treated with the intranasal QER-TFS gel). The respective groups received a daily dosage of normal saline at a rate of 2 mL/kg, oral QER-SOLN at a dose corresponding to 300 mg/kg of QER, and intranasal QER-SOLN gel or QER-TFS gel at a dose equivalent to 10 mg/kg of QER. After 7 days of treatment, the animals were monitored to conduct a behavioral study. A series of behavioral experiments were carried out, including a forced swimming test (FST), tail suspension test (TST), sucrose preference test (SPT), open field test (OFT), interaction test (SIT), and novelty-suppressed feeding (NSF) test. Each rat was then sacrificed, and its brain and nasal mucosa were obtained via decapitation, cleaned with normal saline, and kept at −30 °C for further analysis. Figure 2 depicts the experimental protocol.

##### Behavioral Analysis

Immobility, Swimming, and Climbing in the FST

It is well recognized that antidepressant medication may mitigate immobility [38]. During the whole 6 min testing session of the FST, the durations of immobility, swimming, and climbing activity were manually recorded. We regarded rats to be immobile when it ceased struggling and continued to float without making any movements in the water [39]. After each swim, the water was removed and replaced so there would be no behavioral alterations caused by contamination.

2.Tail Suspension Test (TST)

The rats were subjected to the TST by being hung from their tails using adhesive tape and then held at a height of 50 cm above the surface [40]. The rat was wrapped with a climb stopper to prevent the rat from gripping onto its tail and climbing. Over the course of six minutes, the time that each rat spent immobile and not attempting to escape was recorded and analyzed.

3.Sucrose Preference Test (SPT)

The SPT was carried out following a methodology that was previously published [41]. Each rat received a 1% sucrose solution to drink from two bottles positioned on the opposite sides of the cage for 48 h prior to the SPT. Then, after 14 h without water, we put two pre-weighed bottles in front of each rat, one containing tap water and the other containing the 1% sucrose solution. To eliminate any potential for spatial bias, the placement of the two bottles (left or right) was random. The bottles were measured again after one hour, and the weight difference was used to calculate how much the rats had consumed from each bottle. The preference level for sucrose was determined by calculating the proportion of the 1% sucrose solution ingested with reference to the total amount of liquid added.

4.Open Field Test (OFT)

The OFT was carried out following a methodology that was previously published but with minor adjustments to ascertain the overall locomotor activity of the rats [41]. The experiment was conducted in a square arena measuring 60 cm × 60 cm × 45 cm with white lighting. The floor was split into 16 equal squares. For five minutes, the rat was allowed to wander freely in the center of the open field. Before each round of testing, the equipment was wiped off with a 75% ethanol solution to eliminate any lingering smells. An experienced observer who was ignorant of the experimental design scored the number of crossings (at least three-pawed squares) and rearing encounters (standing on their hind paws) in each recorded test session.

5.Social Interaction Test (SIT)

The SIT involves video recording and analyzing a rat’s behaviors to evaluate active interactions (such as social grooming, crawling under or over, sniffing, boxing, following, chasing, and aggressive behavior), as well as the total number of interactions between a test rat and a new rat [42]. The experiment was carried out in a quiet room with intense white lighting in a chamber measuring 30(L) × 30(W) × 60(H) cm. At the beginning of each round of testing, two weight-matched rats that belonged to the same group were put in opposite corners for a period of 6 min.

6.Novelty-Suppressed Feeding (NSF)

The NSF test followed the previously described procedures [43,44]. This test assesses how long (up to 24 h) it takes rats to approach and consume food in an unfamiliar location after an extended period of food restriction. The time it takes an animal to begin eating reveals how it manages a behavioral issue. The NSF test is utilized for depression-related evaluations because it measures anhedonia in a scenario where there is a conflict between an unfamiliar open space and a food reward, as the capacity to resolve disputes is inversely associated with anxiety and depression [44,45]. The rats were weighed, and all food was removed from their cages, but they continued to have free access to water. Approximately 24 h after the withdrawal of food, the rats were placed in a 40 cm × 60 cm × 50 cm wooden cage that was lit and soundproofed. Each rat was placed in a corner of the testing region, and a small piece of food was put in the middle of the box. The test animal was left alone in a cage with a weighted food piece for 5 min, and the quantity of food ingested was calculated by weighing the food piece at the end of the 5 min period. After examining all the rats in a cage, they were returned to the cage with full access to water and food [46,47].

##### c-fos Immunohistochemistry Analysis

After the FST, the animals were sacrificed, and their brains were removed and stored overnight in fresh paraformaldehyde (4%) [48]. The immunohistochemistry labeling of c-fos protein expression followed the procedure previously described [49]. Brain slices were air-dried at 25 °C overnight before being fixed in paraformaldehyde (4%) for 30 min. The sections were washed twice after fixation and incubated (24 h) at 25 °C with the primary polyclonal rabbit antibody against the c-fos protein. The slices were incubated (2 h) with the secondary anti-rabbit antibody after the sections had been drained and washed twice. Mounted and stained slices were examined via light microscopy. The density of labeled nuclei was estimated within anatomically specified brain areas. Three observations were made for each studied structure in each segment of independently assessed c-fos expression levels in each hemisphere. The observers were unaware of the experiment’s purpose. Th level of expression of the c-fos protein is thought to largely reflect an increase in neuronal activity triggered by a number of physiologically relevant stimuli. This enables the application of immunohistochemical labeling against the c-fos protein to obtain information about neural networks implicated in the investigated effects under various experimental settings.

##### Tolerability and Acute Toxicity Studies

Intranasal histological investigations were carried out to assess the safety of the intranasally administered formulation on the nasal mucosa and rule out any nasal toxicity [27]. After sacrificing the animals, the intranasal mucosa of the control negative, control positive, QER-SOLN gel, and QER-TFS gel groups were surgically removed, decalcified, and preserved in a buffered formalin (10%) solution for subsequent sectioning and analysis with hematoxylin and eosin (H&E) staining under light microscopy [50].

##### Brain Histopathological Examination

The brain tissues from the various groups were embedded in paraffin after being fixed in buffered formalin (10%) and then dissected to produce 4 μm slices. The slices were examined microscopically after being stained with H&E [51,52].

### 2.3. Statistical Analysis

All the experiments were performed in triplicate, and data are presented as means ± SDs. The data for the behavioral analysis were statistically analyzed using a one-way ANOVA with an LSD post-hoc test, using SPSS version 20. A value of *p* < 0.05 was considered statistically significant.

## 3. Results and Discussion

### 3.1. Pharmacodynamic Studies

#### 3.1.1. Behavioral Analysis

Mild depression causes unhappiness and anhedonia, such as decreased food intake and immobility, and a sense of worthlessness, whereas severe depression causes frequent suicide attempts, the main symptom of all types of depression [53]. A behavioral analysis simply tracks the influence of a medicine or treatment on the fundamental symptoms of depression in an animal model. In this study, we employed a behavioral model of depression since it is believed that the most successful method of treating depression is to restore a proper schedule of positive reinforcement for a person by modifying the patient’s environment and/or behavior. Similar to human depression, the animal model exhibited symptoms of chronic moderate stress, social defeat stress, and learned helplessness.

The animal model validity for evaluating depression is established on the basis of observed validity and experience. As previously stated, it is commonly believed that a viable animal model should be able to exhibit anhedonia, a fundamental symptom of depression patients. This attribute is evaluated by measuring the quantity of sugar solution ingested and comparing it to the amount of water consumed [54]. Once behavioral alterations are induced in a rat model, they will last approximately three months, which is the length of time taken by most antidepressants to reverse depression symptoms, such as reduced sucrose consumption [55].

Another symptom that discriminates desperate animals’ behavior from regular animals’ worried behavior is immobility. The FST’s inevitability prompted the rats’ immobility to mirror extinction, miming inhibitory learning. The FST immobility increases cognitive functioning for animal adaptation and survival [56]. The TST was shown to be inappropriate for identifying a "desperate mood" in animals. Instead, it is useful for describing a shift from active to passive behavior as a survival-like response to extreme environmental stress [56]. Therapeutic antidepressants are those that increase escape behavior and reverse immobility.

In the OFT, animals avoid unfamiliar environments by repeatedly circling the outside of a black box and seldom entering its interior. However, their inherent curiosity will prompt them to inspect the center, allowing researchers to spot rodent depression.

Animals with anhedonia, or a reduced ability to experience joy, are commonly assessed using the SPT [57]. An effective model of despair with anhedonia is shown by a decrease in the ratio of sucrose consumed by an experimental animal. The SPT is widely accepted as the most suitable behavioral test for the persistent moderate stress paradigm. In psychopathology, a highly severe depressed state is indicated by two main internal phenotypes: anhedonia and pressure sensitivity. It also reflects tiredness or a lack of energy produced by depressed stress, which antidepressants may alleviate.

The NSF test, which is based on hyponeophagia, provides a sensitive method of assessing the efficacy of antidepressants [44]. The NSF uses the amount of food eaten and the time spent looking for food as its main criteria. NSF drives animals to desire food and still to avoid going towards the center of an open space that has an intense shining light to claim it. These two competing conditions offer a high degree of construct validity for investigating an anxious mood throughout an antidepressant treatment phase [58]. The latency of the NSF test, which is causally connected to the function of hippocampal neurogenesis, allows researchers to explore the relation between a neural circuit and an antidepressant effect [59].

##### Immobility, Climbing, and Swimming in the FST

The FST is the most commonly used test for evaluating the antidepressant efficacy of novel drugs [60]. Figure 3 depicts the outcomes of the FST. The positive control group was the most immobile and spent the least time swimming and climbing. Treatment with QER-SOLN (oral) and QER-SOLN gel (IN) significantly reduced immobility and enhanced swimming and climbing times compared to the depressed positive control (Table 1). However, therapy with QER-TFS gel (IN) dramatically increased climbing and swimming times and decreased immobility compared with all other formulations (Table 1). Several previous investigations have shown that increased climbing and swimming behaviors only happen when an ADD increases the amounts of neurotransmitters, i.e., norepinephrine, serotonin, and dopamine [61].

##### Tail Suspension Test (TST)

The TST showed a significant reduction in immobility for the QER-TFS (IN) gel (27.2 ± 6.1 s), QER-SOLN (oral) (131.8 ± 7.2 s), and QER-SOLN gel (IN) (117 ± 5.8) compared to the positive control group, demonstrating superior antidepressant effectiveness, with the QER-TFS gel (IN) being the most efficient (Figure 4).

##### Sucrose Preference Test (SPT)

All rats in the SPT consumed sucrose, but when the FST provoked a depressed state, a dramatic drop was seen (*p* < 0.05). In contrast, the coadministration of various QER formulations had distinct outcomes. The rats undergoing the SPT showed a significant recovery in sucrose intake (*p* < 0.05) for the QER-TFS (IN) gel (85.5 ± 8.4%), QER-SOLN (oral) (63.2 ± 6.5%), and QER-SOLN gel (IN) (66.2 ± 4.3%) groups compared to the positive control group (33.6 ± 5.1%), as presented in Figure 5.

##### Open Field Test (OFT)

Figure 6 depicts the behavior of the rats in the OFT. The depressed group had markedly reduced cross-lines and rearing numbers than the normal group. The daily treatment of these rats with the QER-TFS gel (IN), QER-SOLN (oral), and QER-SOLN gel (IN) treated the decrease in cross-line numbers existing in the positive control group, as shown in Figure 6. Furthermore, the QER-TFS gel (IN) induced a significant increase in the rearing number (19.7%) compared to the positive control, QER-SOLN gel (IN), and QER-SOLN (oral), as shown in Figure 6.

##### Social Interaction Test (SIT)

In the SIT assessment, depressed behavior was assessed based on interactions (number) and active interactions (duration). The results of the SIT are illustrated in Figure 7. The positive group displayed depressive-like symptoms, as evidenced by a significant decrease in interactions (12.2 + 3.7) and the duration of active interactions (43.3 + 8.6 s) when compared to the normal group (111.6 + 14.6; 61.3 + 7.5). The QER-SOLN (oral) and QER-SOLN gel (IN) significantly improved interactions and active interactions compared with the positive group. Compared to the control positive, the rats administered the QER-TFS gel (IN) exhibited more interactions (59%) and more active interactions (40.7%). Nevertheless, treatment with the QER-TFS gel (IN) resulted in a significant increase in interactions (numbers) and active interaction (sec) as compared to the QER-SOLN gel (IN) and QER-SOLN (oral).

##### Novelty-Suppressed Feeding (NSF)

This test was utilized as an extra measure of depressed behavior [44,62,63]. A longer latency to eat the meal was deemed suggestive of anxious behavior [63]. The one-way ANOVA results show that the latency to feed was significantly lower in the QER-TFS gel (IN), QER-SOLN (oral), and QER-SOLN gel (IN) groups compared to the positive group (Figure 8A). However, treatment with QER-TFS gel (IN) significantly reduced the latency to eating compared with the QER-SOLN (oral) and QER-SOLN gel (IN) groups (Figure 8A). In contrast, there were no significant differences (*p* > 0.05) in the quantity of food consumed across the groups in the NSF test (Figure 8B).

#### 3.1.2. c-fos Immunohistochemistry Analysis

The current study used immunohistochemistry to localize c-fos protein expression as an indication of neuronal activity in order to identify the brain areas involved in behavior modulation during the FST [49]. It is believed that the production of the c-fos protein reflects an increase in neuronal activity induced by a range of physiologically relevant stimuli which may be related to how neuromodulators and neurotransmitters regulate trans-synaptic gene expression. C-fos expression increases in response to visual [64] and auditory [65] stimuli, audiogenic seizures, and the injection of medication [66]. The degree of Fos protein expression indicates the state of treated depression or mood change in control-positive and treated rats. It has been observed that in depressed rodents, performing avoidance or escape tasks enhances Fos-like immunoreactivity in multiple brain areas [67]. Therefore, differences in swim test behavior are explained via the expression of the C-Fos protein in several brain regions subjected to the tension of forced swimming. In depressed rats, increased immobility is related with increased neuronal activity (C-fos expression) in certain brain regions, whereas decreased immobility is connected to decreased neuronal activation in certain brain areas. In other words, during the FST, “passivity” is linked with more neuronal activity than active behavior in certain brain areas [49]. Thus, the brain areas linked to the passive depressed state seem to be revealed via c-Fos expression.

The highest FOS cell numbers (red arrow) were seen in the positive group (Table 2) in the hippocampal region. In addition, a high proportion of FOS cells was also found in the QER-SOLN (oral) and QER-SOLN gel (IN) groups. Interestingly, the group treated with QER-TFS gel (IN) had the lowest FOS cell density.

Figure 9 illustrates the effects of different treatments on FOS expression in brain structures, expressed as positive cell numbers per mm^2^ in normal and depressed rats. In the normal group, the hippocampus showed weak cytoplasmic reactivity (+) for the FOS protein in one cell (DG) and no positive cells in CA3, as depicted in Figure 9A. In addition, the cortical cells of the control negative normal rats demonstrated a mild cytoplasmic reactivity for the FOS protein (++) in the cerebral cortex.

As illustrated in Figure 9B,C, the depressed rats (control positive) had 13 positive cells: 5 with robust cytoplasmic reactivity (+++) for the FOS protein and 8 with mild cytoplasmic reactivity (++). Moreover, the white matter of the depressed rats had 8 positive cells: 4 with robust cytoplasmic reactivity (+++) for the FOS protein and 4 with mild cytoplasmic reactivity (++).

The FOS protein revealed mild cytoplasmic reactivity (++) in four white matter cells of the hippocampus in rats that received QER-SOLN (oral) or the QER-SOLN gel (IN). The QER-SOLN (oral) group showed strong cytoplasmic reactivity (+++) for the FOS protein in six cells (red arrow) and weak reactivity (+) in three cells (Figure 9F). Furthermore, the QER-SOLN gel (IN) group showed weak cytoplasmic reactivity (+) for the FOS protein in five cells (Figure 9E).

Concerning the hippocampus in the QER-TFS gel (IN) group, no positive cells for the FOS protein were found in DGs, as observed in Figure 9D. Three cortical cells from the rats treated with the QER-TFS gel (IN) had mild cytoplasmic reactivity (++) for the FOS protein. Additionally, two cells showed considerable cytoplasmic reactivity (+++) for the FOS protein in the white matter of the rats that received the QER-TFS gel (IN). Overall, in the current investigation, the animals treated with the QER-TFS gel (IN) exhibited the lowest FOS cell density.

#### 3.1.3. Tolerability and Acute Toxicity Studies

An essential criterion for accepting a formulation delivered through the nasal route is evaluating its potential for irritation and toxicity on the delicate nasal mucosa. This was performed by assessing the nasal mucosa of the control negative, control positive, QER-SOLN gel (IN), and QER-TFS gel (IN) groups. Figure 10 depicts the nasal mucosa histology specimens of all four groups. The morphology of the nasal epithelium was normal (black arrow), with average nasal cartilage (red arrow) and average submucosa with average blood vessels (blue arrow), and there was no evidence of inflammation or erosion in the control negative, control positive, and QER-TFS gel (IN) groups, as depicted in Figure 10A–C. In contrast, the QER-SOLN gel (IN) group exhibited pieces of cartilage (red arrow) with ulcerated mucosa (black arrow) and moderate submucosal inflammatory infiltration (blue arrow) (Figure 10D). Overall, the results indicate that the proposed QER-TFS thermosensitive gel (IN) formulation was non-toxic to the nasal mucosa.

#### 3.1.4. Brain Histopathological Examination

Figure 11 and Figure 12 depict a histological study of the structures of the brain. The negative control group had normal brains with typical neurons, meninges, and glial cells (Figure 11 and Figure 12A). In the fibrillary background of the white matter of the normal group, average glial cells and neurons were seen. Additionally, the dentate gyrus, inter-neuron region, pyramidal neurons, granule cells, and blood vessels in the hippocampus of the normal control group were all normal.

Figure 11B,C depict that the control depressed group had congested intra-cerebral blood vessels (yellow arrow), scattered deteriorated neurons with intra-cytoplasmic vacuoles (black arrow), and degenerated glial cells in the fibrillary cell background with considerable micro-cyst development (blue arrow) are also present with an average inter-neuron area (red arrow, B). In contrast, QER-SOLN (oral) group (Figure 11F) showed mildly scattered degenerated pyramidal neurons (yellow arrow) and scattered degenerated granule cells (blue arrow) in (DG), a mildly edematous inter-neuron area (black arrow), scattered neurons with vacuolated cytoplasm (red arrow) in (CA3), and mildly congested blood vessels (green arrow). These degenerative changes were significantly diminished following intranasal therapy in depressed rats (Figure 11D,E), showing normal pyramidal neurons in (CA1) (black arrow), an average inter-neuron area (red arrow), and normal granule cells (blue arrow).

The groups treated with QER-SOLN (oral) showed mildly congested blood vessels, slightly congested blood arteries, average glial cells, scattered, deteriorated neurons, dispersed degraded neurons, and eosinophilic plaque-like regions, as illustrated in Figure 11 and Figure 12F. In addition, the QER-SOLN (oral) hippocampus showed an indistinct CA1 with scattered, degenerated neurons, a mildly edematous inter-neuron area, and an eosinophilic plaque-like area.

For the control positive group (Figure 12B,C) the hippocampus showed a distinct Cornu Ammonis (CA1) with scattered, degenerated pyramidal neurons (black arrow) and granule cells with the inter-neuron area showing markedly congested blood vessels with peri-vascular eosinophilic plaque-like areas (blue arrow) and markedly degenerated pyramidal neurons (black arrow). For the QER-SOLN (oral) group (Figure 12F), the hippocampus showed an indistinct CA1 with scattered, degenerated neurons, a mildly edematous inter-neuron area, and an eosinophilic plaque-like area (blue arrow). These degenerative changes disappeared in the group intranasally administered QER-TFS gel (Figure 12D) and slightly in the QER-SOLN gel (IN) group (Figure 12E).

Overall, the results of the pharmacodynamics, immunohistochemistry, and histopathology experiments reveal that the intranasal delivery of QER through transferosomal nanovesicles based thermosensitive gel is efficient. The improvement in the antidepressant activity of quercetin observed with the intranasally administered nanotransferosomes is attributed to the improved localization of the drug in the brain [33]. The results of this investigation demonstrate that the QER-TFS gel (IN) maintained acceptable levels of QER in the brain, thus reducing the symptoms of depression.

The findings of this study place the intranasally administered QER-TFS gel several degrees above the orally administered QER-SOLN. Intranasal administration opens the door for direct transport to the brain via extracellular and neuronal pathways through the olfactory region, which is termed nose-to-brain (NTB) delivery. The NTB route has been evaluated as a viable way to bypass the BBB and target the brain to treat CNS illnesses promptly and effectively [68]. Therefore, the NTB pathway has been exploited to manage depression symptoms [69,70].

Several studies have shown that brain absorption is restricted if substances undergo metabolism or are effluxed by transporters at the BBB [71]. Accordingly, shielding by nanocarriers may help QER overcome these hurdles and achieve sustained levels inside the brain, a potential reason for the better performance of the QER-TFS gel (IN) compared with the QER-SOLN gel (IN).

Nanotransferosomes promote the delivery of medication to the brain via various pathways, which may account for the increased in vivo antidepressant effectiveness of the QER-TFS gel. The inclusion of the phospholipids bilayer in the vesicle membrane, which creates “new pores” in the paracellular tight junction, contributes to the improved absorption of the TFS [72]. Interestingly, drug delivery systems smaller than 200 nm diameter have demonstrated a high capacity to circumvent the BBB via endocytosis pathways [73,74]. A TFS can effectively transfer the integrated medication to the brain because of its small size and wide surface area, which increases the residence duration and drug levels at the BBB surface [75]. According to some reports, TFS serve as highly flexible penetration enhancers that may open up new holes in the paracellular tight junctions of the nasal membrane, enabling drugs to pass through intact and enhancing drug permeation through the nasal mucosa [28]. In addition, Poloxamer 188, which was used to create the in situ gel, has been reported to assist nanovesicles across the BBB by absorbing lipoproteins from the blood, allowing for receptor-mediated transfer into the brain [76]. Drug transport through the BBB may potentially be aided by surfactant shell-induced membrane fluidization and the inhibition of the efflux transport mechanism [77]. The collaboration of these mechanisms may explain the enhanced therapeutic effectiveness of the QER-TFS gel in an animal depression model.

## 4. Conclusions

In the current investigation, nanotransferosomes improved the delivery of QER to the brain when loaded in a mucoadhesive in situ gel and administered through the nasal route. According to pharmacodynamic, immunohistochemical, and histopathological experiments, the intranasal administration of the QER nanotransferosomal gel substantially alleviated depressive symptoms in rats, warranting a noninvasive therapeutic option for depression. However, extensive preclinical investigations in higher animal models are needed to anticipate its effectiveness in humans.

## Figures and Tables

**Figure 1 pharmaceutics-15-02095-f001:**
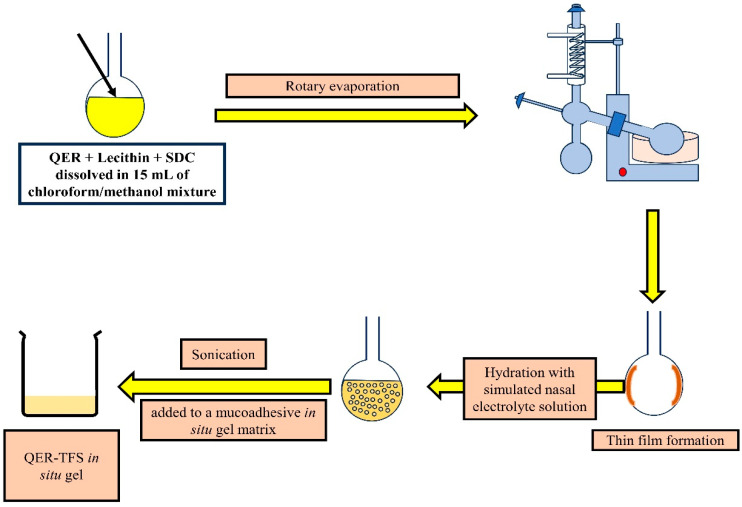
Diagram showing the scheme for formulation of QER-TFS thermosensitive gel.

**Figure 2 pharmaceutics-15-02095-f002:**
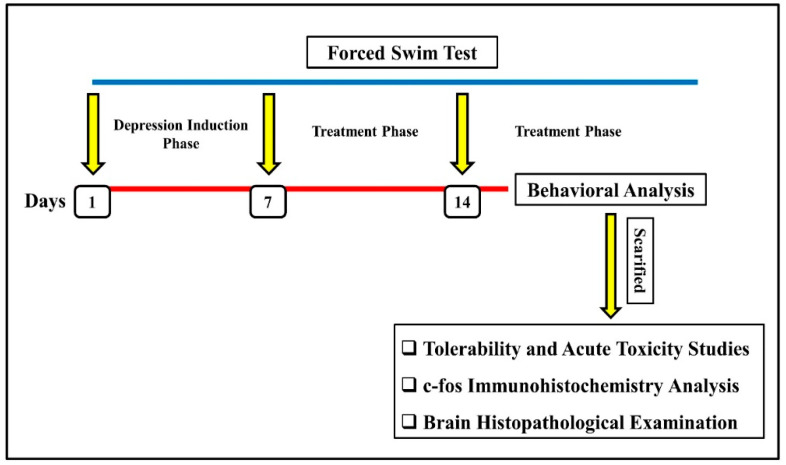
Experimental protocol design.

**Figure 3 pharmaceutics-15-02095-f003:**
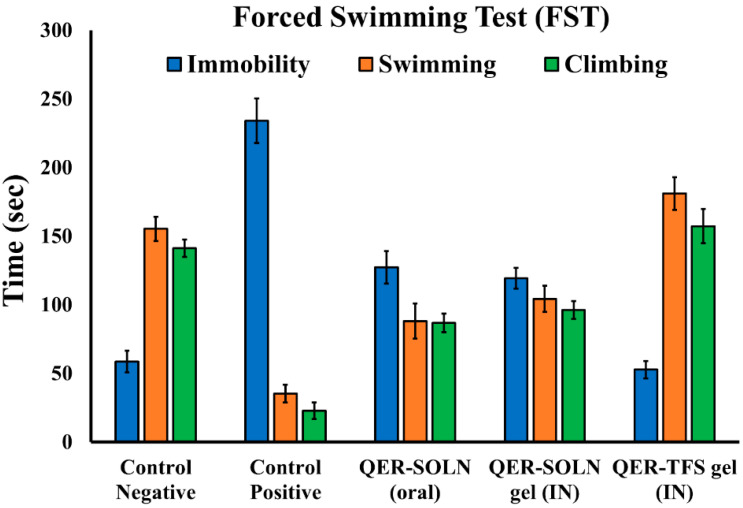
Forced swimming test (FST) results for various formulations (Control Negative, Control Positive, QER-SOLN (oral), QER-SOLN gel (IN), and QER-TFS gel (IN)), showing the effect on climbing, swimming, and immobility times (s).

**Figure 4 pharmaceutics-15-02095-f004:**
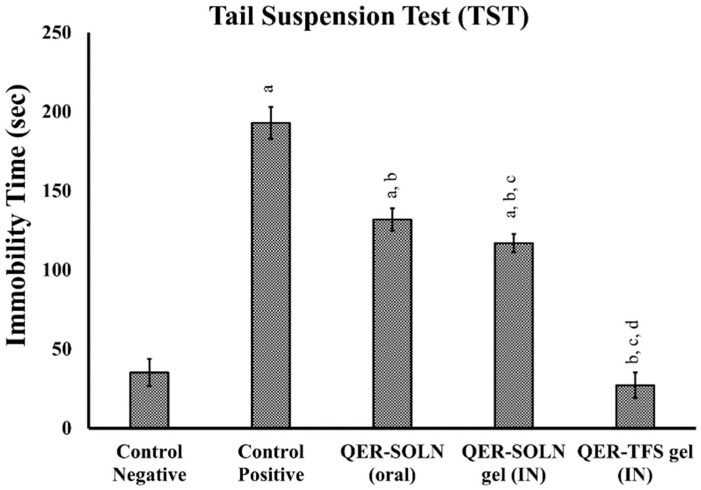
Effects of different formulations (Control Negative, Control Positive, QER-SOLN (oral), QER-SOLN gel (IN), and QER-TFS gel (IN)) on immobility (s) in TST. ^a^
*p* < 0.05 relative to the negative control; ^b^
*p* < 0.05 relative to the positive control; ^c^
*p* < 0.05 relative to QER-SOLN (oral); ^d^
*p* < 0.05 relative to the QER-SOLN gel (IN).

**Figure 5 pharmaceutics-15-02095-f005:**
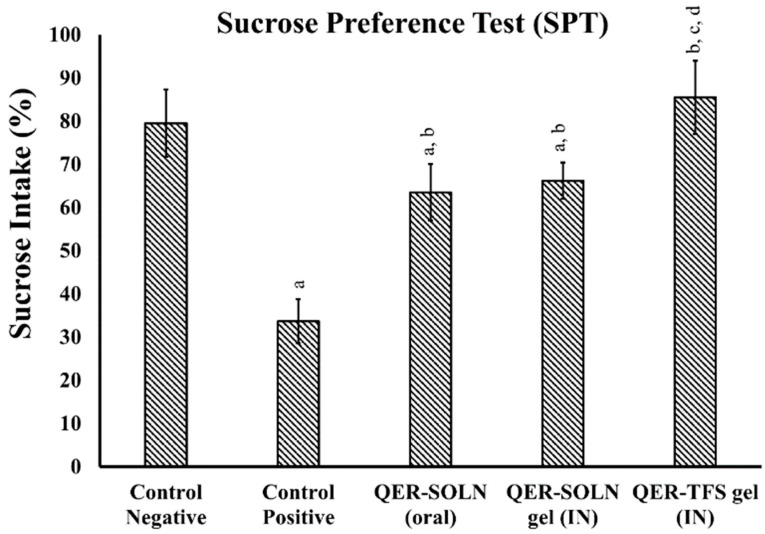
Effects of different formulations (Control Negative, Control Positive, QER-SOLN (oral), QER-SOLN gel (IN), and QER-TFS gel (IN)) on sucrose intake (%) in SPT. ^a^
*p* < 0.05 relative to the negative control; ^b^
*p* < 0.05 relative to the positive control; ^c^
*p* < 0.05 relative to QER-SOLN (oral); ^d^
*p* < 0.05 relative to the QER-SOLN gel (IN).

**Figure 6 pharmaceutics-15-02095-f006:**
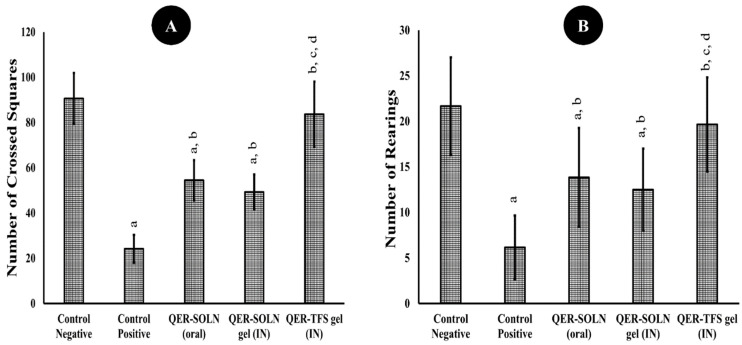
Effects of different formulations (Control Negative, Control Positive, QER-SOLN (oral), QER-SOLN gel (IN), and QER-TFS gel (IN)) on (**A**) the number of crossed squares and (**B**) the number of rearings OFT. ^a^
*p* < 0.05 relative to the negative control; ^b^
*p* < 0.05 relative to the positive control; ^c^
*p* < 0.05 relative to QER-SOLN (oral); ^d^
*p* < 0.05 relative to the QER-SOLN gel (IN).

**Figure 7 pharmaceutics-15-02095-f007:**
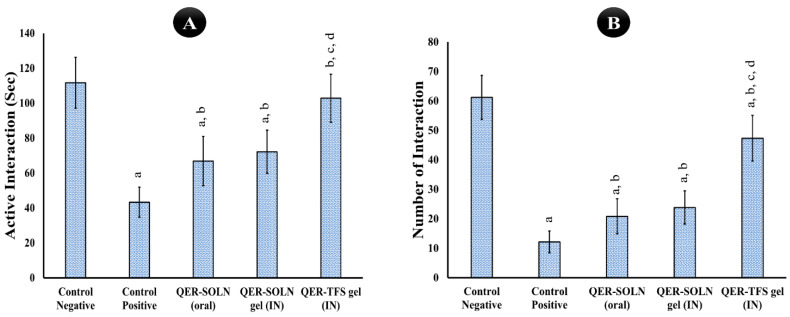
Effects of different formulations (Control Negative, Control Positive, QER-SOLN (oral), QER-SOLN gel (IN), and QER-TFS gel (IN)) on (**A**) active interaction time (sec) and (**B**) number of interactions in a social interaction test (SIT). ^a^
*p* < 0.05 relative to the negative control; ^b^
*p* < 0.05 relative to the positive control; ^c^
*p* < 0.05 relative to QER-SOLN (oral); ^d^
*p* < 0.05 relative to the QER-SOLN gel (IN).

**Figure 8 pharmaceutics-15-02095-f008:**
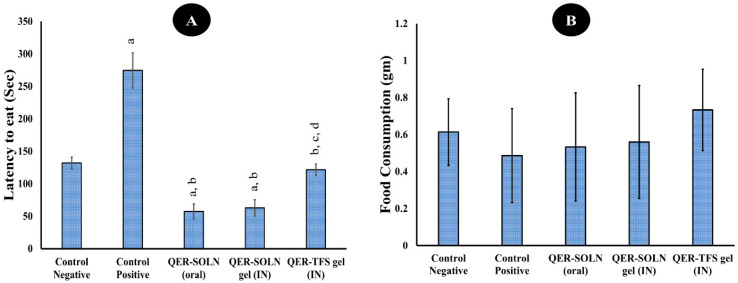
Effects of different formulations (Control Negative, Control Positive, QER-SOLN (oral), QER-SOLN gel (IN), and QER-TFS gel (IN)) on (**A**) latency to feed (s) and (**B**) food consumption (gm) in a novelty-suppressed feeding test (NSF). ^a^
*p* < 0.05 relative to the negative control; ^b^
*p* < 0.05 relative to the positive control; ^c^
*p* < 0.05 relative to QER-SOLN (oral); ^d^
*p* < 0.05 relative to the QER-SOLN gel (IN).

**Figure 9 pharmaceutics-15-02095-f009:**
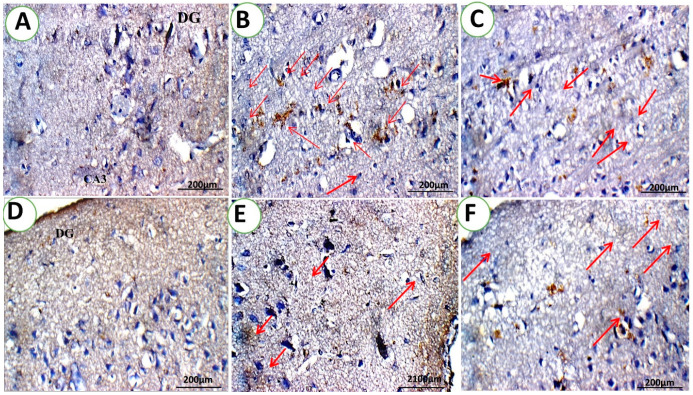
Photomicrographs show the effects of different treatments on c-fos expression in the hippocampus area (red arrow) in normal and depressed rats. (**A**) Control negative group; (**B**,**C**) control positive group; (**D**) QER-TFS gel (IN) group; (**E**) QER-SOLN gel (IN) group; and (**F**) QER-SOLN (oral) group. All images are stained with hematoxylin–eosin and ×400 magnification.

**Figure 10 pharmaceutics-15-02095-f010:**
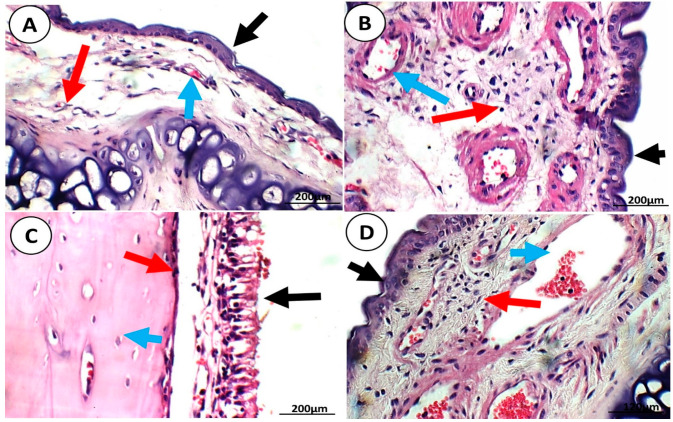
Light photomicrographs showing the nasal epithelia of the (**A**) negative control group; (**B**) positive control group; (**C**) QER-TFS gel (IN) group; and (**D**) QER-SOLN gel (IN) group. In the control negative, control positive, and QER-TFS gel (IN) groups, the nasal walls showed average intact epithelial linings (black arrow), average nasal cartilage (red arrow), and average submucosa with average blood vessels (blue arrow). Scale bar: 200 µm. In the QER-SOLN gel (IN) group, pieces of cartilage (red arrow) with ulcerated mucosa (black arrow) and moderate submucosal inflammatory infiltration (blue arrow) are visible. All images are stained with hematoxylin–eosin and shown at ×400 magnification.

**Figure 11 pharmaceutics-15-02095-f011:**
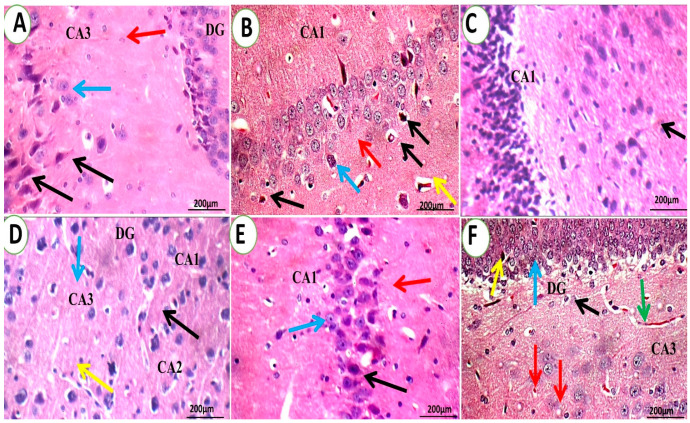
Photomicrographs show the effects of different treatments on different areas of brain histopathology. (**A**) Control negative group, (**B**,**C**) control positive group, (**D**) QER-TFS gel (IN) group, (**E**) QER-SOLN gel (IN) group, and (**F**) QER-SOLN (oral) group. In the control positive group (**B**,**C**), congested intra-cerebral blood vessels (yellow arrow), scattered deteriorated neurons with intra-cytoplasmic vacuoles (black arrow), and degenerated glial cells in the fibrillary cell background with considerable micro-cyst development (blue arrow) are also present with an average inter-neuron area (red arrow, **B**). The QER-SOLN (oral) group (**F**) showed mildly scattered degenerated pyramidal neurons (yellow arrow) and scattered degenerated granule cells (blue arrow) in (DG), a mildly edematous inter-neuron area (black arrow), scattered neurons with vacuolated cytoplasm (red arrow) in (CA3), and mildly congested blood vessels (green arrow). These degenerative changes were significantly diminished following intranasal therapy in the depressed rats (**D**,**E**), showing normal pyramidal neurons in (CA1) (black arrow), an average inter-neuron area (red arrow), and normal granule cells (blue arrow). All images are stained with hematoxylin–eosin and shown at ×400 magnification.

**Figure 12 pharmaceutics-15-02095-f012:**
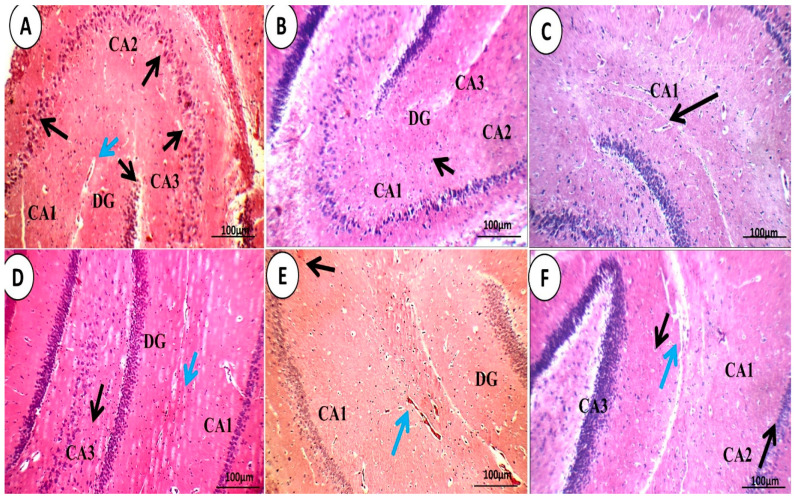
Photomicrographs showing the effects of different treatments on the histopathology of the hippocampus. (**A**) Control negative group, (**B**,**C**) control positive group, (**D**) QER-TFS gel (IN) group, (**E**) QER-SOLN gel (IN) group, and (**F**) QER-SOLN (oral) group. The hippocampus of the control positive group (B&C) showed distinct a Cornu Ammonis (CA1) with degenerated, scattered pyramidal neurons (black arrow) and granule cells with the inter-neuron area showing markedly congested blood vessels with peri-vascular eosinophilic plaque-like areas (blue arrow), markedly degenerated pyramidal neurons (black arrow) (H&E × 400) in (C) the positive group. In the QER-SOLN (oral) group, the hippocampus (**F**) showed indistinct CA1 with scattered, degenerated neurons, a mildly edematous inter-neuron area, and an eosinophilic plaque-like area (blue arrow). These degenerative changes disappeared in the group intranasally administered QER-TFS gel (**D**) and slightly in the QER-SOLN gel (IN) group (**E**). All images are stained with hematoxylin–eosin and are shown at ×200 magnification.

**Table 1 pharmaceutics-15-02095-t001:** Effects of different formulations (Control Negative, Control Positive, QER-SOLN (oral), QER-SOLN gel (IN), and QER-TFS gel (IN)) on the modified FST in depressed rats. ^a^
*p* < 0.05 compared with the negative control group; ^b^
*p* < 0.05 compared with the positive control group; ^c^
*p* < 0.05 as compared with the QER-SOLN (oral) group; ^d^
*p* < 0.05 compared with the QER-SOLN gel (IN).

Group	Treatment	Dose	Parameters		
			Immobility (s)	Swimming (s)	Climbing (s)
1	Control Negative	2 mL/kg (normal saline, not depressed)	58.7 ± 7.8	155.3 ± 8.7	141.2 ± 6.3
2	Control Positive	No treatment, depressed	234.2 ± 16.1 ^(a)^	35.6 ± 6.5 ^(a)^	22.8 ± 6.0 ^(a)^
3	QER-SOLN (oral)	300 mg/kg (quercetin, depressed)	127.3 ± 11.8 ^(a,b)^	88.2 ± 12.8 ^(a,b)^	86.8 ± 6.7 ^(a,b)^
4	QER-SOLN gel (IN)	10 mg/kg (quercetin, depressed)	119.4 ± 7.5 ^(a,b)^	104.3 ± 9.5 ^(a,b,c)^	96.2 ± 6.4 ^(a,b)^
5	QER-TFS gel (IN)	10 mg/kg (quercetin, depressed)	52.8 ± 6.4 ^(b,c,d)^	181.2 ± 11.9 ^(a,b,c,d)^	157.3 ± 12.5 ^(a,b,c,d)^

**Table 2 pharmaceutics-15-02095-t002:** Effects of different treatments (Control Negative, Control Positive, QER-SOLN (oral), QER-SOLN gel (IN), and QER-TFS gel (IN)) on the neurological activity and positive immune cells (%) in the brain after the FST.

Group	Treatment	Positive Immune Cells (%) *
**1**	Control Negative (normal)	15.00 ± 3.00
**2**	Control Positive (depressed)	85.00 ± 1.00
**3**	QER-SOLN (oral)	75.00 ± 3.00
**4**	QER-SOLN gel (IN)	67.00 ± 3.00
**5**	QER-TFS gel (IN)	5.00 ± 1.00

* the values are means ± SDs (*n* = 6).

## Data Availability

Data sharing is contained in this article.

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
