# Peer review of "Intranasal Nanotransferosomal Gel for Quercetin Brain Targeting: II. Antidepressant Effect in an Experimental Animal Model"

_pharmaceutics, 2023, doi:10.3390/pharmaceutics15082095_

Round 1

Reviewer 1 Report (Previous Reviewer 1)

The author should clairfy the following points 

1. Why in the figure 8,9,10 the image magnification is different however scare bar is the same 

2. Clarify the loading % in the mucoadhesive gel which authors have used for the experiment.

3. Author needs to characterize the hydrogels using different parameters 

Rheological assessment, release behaviour, swelling, degradation etc.

4. Author should cite the following article in the introduction Bioavailability enhancement techniques for poorly aqueous soluble drugs and therapeutics. Biomedicines10(9), 2055.

5.

In-vitro drug release behaviour swelling rheological behaviours of the hydrogels Q

NA

Author Response

  • Authors: The authors would like to express their gratitude to the reviewer for his insightful remarks. Your expertise is much appreciated.

  • The author should clarify the following points. 
  • Why in figures 8,9,10 the image magnification is different however scare bar is the same.
  • Authors: Thank you for the in-depth analysis. We already indicated in the legends of Figures 8, 9, 10 that all images were recorded at the same magnification, X400. Anyway, the scale bars of the images of these figures were reviewed and edited.
  • Clarify the loading % in the mucoadhesive gel which authors have used for the experiment.
  • Authors: The loading percent of the drug incorporated into the mucoadhesive gel is now included in the manuscript. Thank you for your suggestion.
  • Author needs to characterize the hydrogels using different parameters Rheological assessment, release behavior, swelling, degradation etc.
  • Authors: The optimum formula was incorporated into a gel matrix that was fully assessed in a previous study (DOI: https://doi.org/10.2147/IJN.S381353). Concerning the release behavior, it was evaluated in part I of our work (DOI: https://doi.org/10.3390/pharmaceutics15071805). Additionally, ex-vivo permeation, stability, and pH were also evaluated in part I. Thank you so much for the comprehensive review.
  • Author should cite the following article in the introduction Bioavailability enhancement techniques for poorly aqueous soluble drugs and therapeutics. Biomedicines10(9), 2055.
  • Authors: The article “Bioavailability enhancement techniques for poorly aqueous soluble drugs and therapeutics. Biomedicines10(9), 2055” is now cited in the introduction section. Thank you. 

Reviewer 2 Report (Previous Reviewer 2)

The authors responded appropriately to previous comments resulting in an improved manuscript. There are some minor text formatting suggestions that could be improved. Namely spacing in the tolerability and acute toxicity studies section within methods.

Author Response

  • The authors responded appropriately to previous comments resulting in an improved manuscript. There are some minor text formatting suggestions that could be improved. Namely spacing in the tolerability and acute toxicity studies section within methods.
  • Authors: The authors would like to express their gratitude to the reviewer for his positive comments. Your expertise is much appreciated. The font and spacing in the tolerability and acute toxicity studies section within methods are now edited in the manuscript. Thank you so much.

Round 2

Reviewer 1 Report (Previous Reviewer 1)

Most of the points have been revised by the authors 

NA

Author Response

  • Most of the points have been revised by the authors.
  • Authors: The authors would like to express their gratitude to the reviewer for his positive comments. Your expertise is much appreciated.

This manuscript is a resubmission of an earlier submission. The following is a list of the peer review reports and author responses from that submission.

Round 1

Reviewer 1 Report

The author should revise the manuscript as following suggested points

1. The author should draw a schematic illustration that can explain the whole store to the readers at a glance. The illustration should keep as Figure 1 including the formulation image in this figure.

2. Statistical analysis represent by proper with (*,**,***) not with alphabets and clarly

illustrate from which group to which group compared.

3. Include a supportive video or images for the behavioural Immobility

, Swimming, and Climbing results only numerical values data is not reliable.

4 In the discussion part explain the possible mechanism of action with proper illustration and pathways clearly.

5. in all histopathology images include the scale bar.

6. Organise the data in a good manner so that reader can understand without confusion.

NA

Reviewer 2 Report

The authors describe the use of use of natural product Quercetin for use as an antidepressant therapeutic in a nasal formulation. The authors use a forced swim test model of depression, which is a well-established disease model. The authors employ an untreated control group, a control group with the induced phenotype, an orally dosed group, and two nasally dosed groups, one with the TFS gel formulation. The results show a return to baseline in many of the behavioral tests with little negative impact of the treatment. The manuscript suffers from a lack of exterior comparison of their results making it difficult to determine the significance of the work.

Major comments:

There’s no comparison of this therapeutic regime in comparison to existing therapeutic options. While the authors did include both positive and negative controls this does not explain how the nasal delivery of their therapeutic compares with current standard of care.

Behavioral despair is the only depression model system used. It would be beneficial to consider how these results may translate to other models of depression or in the clinic.

All of the images contain scalebars but there is no information on the size of these scale bars. Images appear to be at different magnification throughout making it difficult to discern details of features or structures. Images are irregularly labeled with the regions and structures of the brain that the text describes.

Quantification of FOS cells appears in the text but it is unclear how representative these numbers are. How many animals? How large of an area. Additionally cells with weak, moderate, and robust expression are quantified, but there is no clear rationale for this distinction.

Statements in the text are not backed up by the data. Specifically the text states: :” However, treatment with QER-TFS gel (IN) significantly reduced the latency to eating compared with QER-SOLN (oral) and QER-SOLN gel (IN) groups.” Which is unsupported by the data.

There is little to no discussion of how the presented results fit into the larger field of ADDs, nor to how this gel composition and nasal delivery system compares to other. SIfnificantly more discussion and included.

In the methods section the authors state: “Preparation method of QER-TFS was detailed in the first part” it is unclear what the authors are referring to.

Minor comments

There were multiple fonts, size, and styles used through out the text inconsistently.

Figures 6 and 7 were distorted

Figure 6 caption appears twice

Abbreviations for the behavioral tests were used before they were introduced.

Brain sectioning was described for histopathological examination but not for IHC

Naming of the control groups was confusing, control positive and control negative were not intuitive names.

Language was generally non-problematic, though as defined ADD is a plural noun and was not used as such.